# Design and Optimization of Orally Administered Luteolin Nanoethosomes to Enhance Its Anti-Tumor Activity against Hepatocellular Carcinoma

**DOI:** 10.3390/pharmaceutics13050648

**Published:** 2021-05-02

**Authors:** Mahmoud M. A. Elsayed, Tarek M. Okda, Gamal M. K. Atwa, Gamal A. Omran, Atef E. Abd Elbaky, Abd El hakim Ramadan

**Affiliations:** 1Department of Pharmaceutics and Clinical Pharmacy, Faculty of Pharmacy, Sohag University, P.O. Box 82524, Sohag 82524, Egypt; 2Department of Biochemistry, Faculty of Pharmacy, Damanhur University, Damanhur 22516, Egypt; Tarekokda@pharm.dmu.edu.eg (T.M.O.); Gamal.omran@pharm.dmu.edu.eg (G.A.O.); 3Department of Biochemistry, Faculty of Pharmacy, Port Said University, Port Said 42515, Egypt; Gamal.Mohamed@pharm.psu.edu.eg (G.M.K.A.); dean@pharm.psu.edu.eg (A.E.A.E.); 4Department of Pharmaceutics and Industrial Pharmacy, Faculty of Pharmacy, Port Said University, Port Said 42515, Egypt; Abd.Elhakim.Ramadan@pharm.psu.edu.eg

**Keywords:** luteolin, ethosomes, nano-sized vesicles, nanoparticle, hepatocellular carcinoma, oxidative stress biomarkers

## Abstract

Luteolin (LUT) is a natural flavonoid with low oral bioavailability with restricted clinical applications due to its low solubility. LUT shows significant anti-tumor activity in many cancer cells, including hepatocellular carcinoma (HCC). The most recent trend in pharmaceutical innovations is the application of phospholipid vesicles to improve the solubility of such hydrophobic drugs. Ethosomes are one of the most powerful phospholipid vesicles used to achieve that that target. In this study, LUT-loaded ethosomal nanoparticles (LUT-ENPs) were prepared by the cold method. Full factorial design and response surface methodology were used to analyze and optimize the selected formulation variables. Drug entrapment efficiency, vesicle size, zeta potential, Fourier transform infra-red spectroscopy, scanning electron microscopy, and cumulative percent drug released was estimated. The selected LUT-ENPs were subjected to further investigations as estimation of hepatic gene expression levels of GPC3, liver biomarkers, and oxidative stress biomarkers. The prepared LUT-ENPs were semi-spherical in shape with high entrapment efficiency. The prepared LUT-ENPs have a small particle size with high zeta potential values. The in vitro liver biomarkers assay revealed a significant decrease in the hepatic tissue nitric oxide (NO), malondialdehyde (MDA) content, and the expression of the GPC3 gene. Results showed a high increase in the hepatic tissue levels of glutathione (GSH) and superoxide dismutase (SOD). Histopathological examination showed a small number of hepatic adenomas and a significant decrease of neoplastic hepatic lesions after treatment with LUT-ENPs. Our results firmly suggest the distinctive anti-proliferative activity of LUT-ENPs as an oral drug delivery system for the treatment of HCC.

## 1. Introduction

Based on the previous literature, liver cancer remains the leading cause of cancer-related deaths worldwide despite, advances in avoidance strategies, yearly checking, and advanced innovations in diagnosis and anticipation [1]. There are two stages of liver cancer, primary and secondary. Hepatocellular carcinoma (HCC) and cholangiocarcinoma are two forms of primary liver cancer that start in the liver tissue (cancer of the bile ducts). As cancer spreads from other parts of the body to the liver, it is known as secondary metastatic liver cancer [2,3]. The most prevalent type of liver cancer is HCC, which accounts for nearly 90% of all liver cancer [2,4,5]. Chemotherapy and other traditional cancer therapies utilized in the treatment of HCC are restricted due to their adverse effects [4,5]. Medicinal plants have great importance due to their [6] antioxidant, antitumor activities, a promising growth inhibitory effect against the skin, and ovarian cancer [7]. Medicinal plants are used for cancer treatment because of their ability to inhibit or delay cancer proliferation. Additionally, they could boost the immune system and physiological status. Additionally, they provide a great alternative and/or an adjuvant option to a chemotherapeutic agent by reducing or preventing their side effects [8]. Curcumin, Resveratrol, Genistein, Epigallocatechin gallates (EGCG), and other phytochemicals have been shown to have anti-cancer properties in breast and prostate cancers [9]. Luteolin (LUT) is a flavonoid that can be found in a variety of plants, including fruits, vegetables, and medicinal herbs [8]. In several human cancer cells, LUT induces cell cycle arrest and/or apoptosis [10]. LUT induces cytotoxicity in cancer cells by inhibiting cell-cycle pathways and encouraging apoptosis pathways. Moreover, it inhibits cell survival pathways, represses apoptosis inhibitors, and blocks anti-apoptotic Bcl-2 family members [11]. Because of its low solubility, LUT has a restricted oral bioavailability [9]. The oral pharmaceutical formulation of hydrophobic drugs suffers from many difficulties that result in variation in its bioavailability, due to its hydrophobic nature [12,13,14]. As a result, many nanotechnological approaches have been developed for enhancing the solubility and bioavailability of poorly water-soluble drugs including ethosomes, niosomes, liposomes, micelles, conjugates, nanoparticles, and nano globules [15]. Ethosomes are soft, malleable vesicles that serve as non-invasive delivery vehicles for a variety of medications. They are primarily composed of phospholipids, water, and ethanol (EtOH) [16]. Ethosomes help hydrophobic drugs dissolve more easily [16]. Ethosomes are similar to liposomes; however, since alcohol increases the solubility of fat-soluble compounds, ethosomes have high drug entrapment efficiency (EE%) for hydrophobic compounds [11]. Ethosomes are mostly used in transdermal drug delivery systems, but they can also be used to distribute poorly water-soluble drugs orally [17,18]. The incorporation of LUT in ethosomal formulations is predicted to overcome the poor water solubility and enhance its bioavailability.

In this study, we used ethosomal nanoparticles (ENPs) as a drug procurement route for LUT to increase its bioavailability and anti-cancer activity. As far as we know, this is the first study to report on the formulation of orally administered LUT-ENPs for use in the treatment of HCC. The size, morphology, FTIR, and in vitro release of the prepared ENPs were all measured. Furthermore, the estimation of oxidative stress and liver biomarkers in vivo.

## 2. Materials and Methods

### 2.1. Materials

Luteolin, soy lecithin (SL), cholesterol, and 12,000 molecular cuts off cellophane membrane were obtained from Sigma Chemical Co. (St. Louis, MO, USA). Propylene glycol (PG), EtOH, potassium di-hydrogen orthophosphate anhydrous, and di-potassium hydrogen orthophosphate anhydrous, were purchased from Elnasr pharmaceutical chemical Co. (Cairo, Egypt). Diethylnitrosamine (DENA) and carbon tetrachloride (CCL_4_) were obtained from Sigma Aldrich (USA). Hydrophobic filter, membrane filter, diameter pore 0.2 µm was obtained from Versapor, German Sciences (Primo Heraeus, Hanau, Germany). All other reagents used were of analytical grade.

### 2.2. Animals

The rats in this study were 150 g, 15 weeks old male Wistar rats held in regular housing conditions with free access to a standard pellet diet and tap water (22 ± 2 °C temperature, 50 ± 5% humidity, 12/12 h light/dark cycle). The animals were obtained from the animal house of the Faculty of Medicine, Mansoura University, Egypt. The Guidelines of the National Institute of Health approved by the animal ethics committee of Faculty of Pharmacy, Damanhur University (No. 117B24, 16 May, 2018) was followed for the care and use of the rats.

### 2.3. Methods

#### 2.3.1. Determination λ_max_ of LUT

Ten milligrams of LUT were dissolved in one hundred milliliters of ethyl alcohol to make a 100 µg/mL stock solution. From this solution, ten milliliters were diluted to one hundred milliliters with phosphate buffer saline (PBS) pH 7.4. Double beam spectrophotometer, UV-1601, Shimadzu Co., Tokyo, Japan was used to fully scan this solution to decide the λ_max_ of LUT in the UV region 200–800 nm.

#### 2.3.2. Validation of the Analytic Technique

According to the International Council for Harmonisation (ICH) guidelines, the method of the assay was validated regarding the linearity, limit of detection (LOD), and limit of quantitation (LOQ), accuracy, and precision.

Three replicates of nine concentrations from the drug solution were used to test linearity and construct the calibration curve. The slope of the calibration curve and the standard deviation of response was used to calculate the LOD and LOQ using the following equations; LOD = 3.3 σ/S and LOQ = 10 σ/S (σ represent the intercept standard deviation, S represent the calibration slope). A five concentration within the specific range (*n* = 3) was used to test accuracy, the results of measurements are presented as mean ± standard deviation. Intra-day and inter-day precision were evaluated, the intra-day precision was estimated by replicating during the same day three concentrations, three times. The inter-day precision by replicating during three successive days, at three concentrations, three times. Analytical solution stability was evaluated by dissolving LUT in (1:3) EtOH: PBS, pH 7.4 and transferred to sealed tubes, protected from light and stored at room temperature, and 5 °C.

#### 2.3.3. Preparation of LUT-ENPs

LUT-ENPs were prepared by the cold method [17], where LUT, soy lecithin (SL), and cholesterol in different ratios as shown in Table 1, were dissolved in EtOH. EtOH solution was stirred using a magnetic stirrer at 1500 rpm and 40 °C for 30 min. Stirring was continued for another five minutes after the addition of PG. In a separate vessel, water was heated to 40 °C. Once both mixtures reached 40 °C, water was added at a constant rate of 1 mL/min through a syringe pump. After the complete addition of water, stirring was continued for an additional 30 min. The final milky dispersion of LUT loaded ethosomes was left to cool at room temperature, and then left in the refrigerator until further investigations (Figure 1) [16,17].

For the optimization of different formulation variables three levels of each independent variable were used. The selected levels of EtOH concentrations (X_1_) were 15(−1), 30(0), and 45% (1) (*v/v*), whereas the selected levels of SL concentrations were (X_2_) 2(−1), 4(0), and 6(1) %(*w/v*) (Table 1) [19,20].

#### 2.3.4. Characterization of the Prepared LUT-ENPs

##### Determination of Entrapment Efficiency Percentage (EE%)

The ultracentrifugation method was used to characterize the EE% of LUT-ENPs using the Biofuge Centrifuge (Primo Heraeus, Hanau, Germany) at 10,000 rpm for 60 min [19,21]. PBS, pH 7.4 was used to dilute 1 mL of the supernatant obtained and measured at 356 nm using PBS, pH 7.4 as a blank. The EE% was calculated using Equation (1).
EE% = [(E_t_ − E_f_)/E_t_] × 100(1)
where E_t_ is the total amount of the drug added and E_f_ is the amount of free drug.

##### Morphology of The Prepared LUT-ENPs

LUT-ENPs inspected by scanning electron microscopy (Jeol JSM-5400 LV, Jeol Ltd., Tokyo, Japan), Drops from the preparation were spread over a twofold adhesive layer of carbon on a metal bolster and covered with gold using an Ion-Sputtering gadget (Jeol Fine-Coat JFC 1100E, Jeol Ltd., Tokyo, Japan) [22].

##### LUT-ENPs Size and Polydispersity Index (PDI)

LUT-ENPs vesicle size was estimated using Malvern Zetasizer 300 HAS (Malvern Instrument, Worcestershire, UK) at 25 °C. Milli-Q-water was used to dilute LUT-ENPs to supply an appropriate scattering intensity. Tests were inspected three times and the average was obtained. PDI was moreover examined to indicate the homogeneity of the particle size.

##### Zeta Potential Determination

A photon correlation spectroscopy (Zetasizer Nano ZS, ZEN 3600; Malvern Instruments, Malvern, UK) was used to estimate the Zeta potential of LUT-ENPs. Six measurements were estimated, and the mean was used ±SD.

##### Fourier Transform Infrared Spectroscopy (FTIR)

FTIR study was conducted to investigate any interaction between the drug and the other ingredients used in the formulations. Excipients (individually), LUT, LUT-ENPs, physical mixture corresponding to LUT-ENPs were examined by Fourier Transform (IR-476-Shimadzu Kyoto, Tokyo, Japan) in the region from 4000 to 400 cm^−1^ using the KBr disc method [23,24]. The samples were compressed by a Shimadzu SSP-CoA IR compression machine at a pressure of 6 ton/cm^2^ after trituration and complete blending with KBr. The spectra were conducted at 2 cm^−1^, after charging the plates within the instrument light path.

#### 2.3.5. In Vitro LUT-ENPs Release Study

Five milliliters of ethosomal suspension free from unentrapped LUT (equivalent to 5 mg drug) were charged into a semipermeable cellophane membrane dialysis bag having 12,000 molecular weight cut off the range, tied from both ends with cotton threads, and placed in 250 mL PBS, pH 7.4. The system was maintained at 37 °C with shaking at 100 rpm in a thermo-controlled shaker. At a specified time interval for 48 h, 5 mL of the dialysate were withdrawn and replaced with the same volume of fresh PBS, pH 7.4 to maintain the sink condition [18]. The amount of LUT released was measured spectrophotometrically at 356 nm using PBS, pH 7.4 as blank and the average of three measurements was used [25,26].

#### 2.3.6. Kinetic Studies

To obtain the most appropriate release mechanism of LUT from the prepared LUT-ENPs, zero-order, first-order, second, and Higuchi-diffusion models [2,14,27] were used. The best-fitted model would describe the mechanism of drug release.

#### 2.3.7. Vesicle Stability Studies

The selected LUT-ENPs formula (F8) was put away in vials (borosilicate, clean, dry, airproof, and dark-colored) and subjected to stability study at 4 ± 0.5 °C and 25 ± 2 °C for 12 weeks. The size of the vesicle, EE%, and percentage of LUT release after 12 h was investigated and compared to the same properties of the tested formula at time zero [28,29]. Every 4 weeks, samples were withdrawn from the tested preparations. Each test was repeated three times and the mean was obtained [27].

#### 2.3.8. In Vivo Study

Our animal model was optimized concurring with the strategy of Pandurangan et al., 2014 [30]. Rats (90, male and healthy) were partitioned haphazardly to give six bunches (each bunch contains 15 rats). The negative control bunches (bunch A and bunch B) were received a solution of CMC-Na in water (0.5% *w/v*) and blank ethosomes, separately every day by verbal gavage along the test period. A single I.P injection dose of DENA (200 mg/kg) [31], Recently dissolved in sterile 0.9% saline solution was administered to bunches C, D, E, and F. Two weeks later, each animal in the last 4 bunches were received an S.C injection of CCl_4_ (3 mL/kg/week) for 6 weeks to enhance the carcinogenic effect of DENA [32]. The positive control bunches (bunch C and bunch D) were received a solution of sodium caboxymethyl cellulose (CMC-Na) in water (0.5% *w/v*) and blank ethosomes, individually every day by verbal gavage during the test. Bunch E has daily received a dose of 100 mg/kg, for four weeks from 10 mg/mL LUT suspended in a solution of CMC-Na in water (0.5% *w/v*) by verbal gavage [33]. Bunch F was received the same dose amount (10 mg/mL) from LUT-ENPs as shown in bunch E. In more concise word each rat 150 g in weight was received 1.5 mL enclosing 15 mg LUT.

After 12 h from the last treatment dose, sampled blood was collected under ketamine anesthesia (100 mg/kg) by cardiac puncture method, wait for 20 min till coagulation. Centrifugation for 15 min of the blood tests at 4000 rpm to recover the serum at 4 °C employing a cooling Beckman show centrifuge (L3-50, Waltham, MA, USA) and put away at −40 °C till assessment of liver biomarkers. Livers were removed after cervical euthanization, washed with low-temperature PBS, dried with tissue paper, stored in formalin (10% for histopathological examination [34].

#### 2.3.9. Detection of Glypican 3 by Real-Time Polymerase Chain Reaction (RT-PCR)

One section of the liver was directly harvested using the lysis buffer presented in the GF-1 total RNA extraction kit (Vivantis Technologies, Selangor, Malaysia) according to the manufacturer’s instructions, to achieve maximum yield of intact RNA. All steps of total RNA extraction were accomplished on ice using ice-cold reagents. RNA concentrations were determined using the SPECTROstar Nano-spectrometer (BMG Labtech, Bad Friedrichshall, Germany). RNA quality was determined by measuring the 260/280 ratio. Single-stranded cDNA was created from 2 μg total RNA using the cDNA synthesis kit supplied in the 2-step RT-PCR kit (Vivantis Technologies) following the manufacturer’s guidelines. The obtained cDNA was applied to quantify target mRNA expression using RT-PCR amplification (Fisher Scientific Ltd., Vantaa, Finland) with a total reaction volume of 20 μL per well of an RT-PCR plate. The amplification reaction mixture contained 2 μL cDNA, 0.3 μL of each primer (10 μm), 10 μL SYBR green universal master mix (Thermo Fisher Scientific, Waltham, MA, USA), and 7.4 μL DNase-free water. The RT-PCR reaction conditions were as follows: 35 cycles of initial denaturation (95 °C, 3 min), denaturation (95 °C, 30 s), annealing (temperature-dependent on a specific gene, 30 s), and extension (72 °C, 30 s); followed by the final extension (72 °C, 5 min). The primers were obtained from Vivantis Technologies (Selangor, Malaysia) and were designed using PubMed and tested for annealing temperatures.

GPC3 forward Tm = 57.79 °C: 5′-GTGCTGGAACGGACAAGAG-3′

GPC3 reverse Tm = 58.05 °C: 5′-TTCTTCATCCCATTCCTTGC-3′

β-actin forward Tm = 60 °C: 5′-CTAAGGCCAACCGTGAAAAG-3′

β-actin reverse Tm = 60 °C: 5′-TACATGGCTGGGGTGTTGA-3′

The RT-PCR data were estimated to calculate fold changes and relative expression using the 2^−^^△△CT^ method by Livak [35]. β-actin was used as the endogenous reference gene.

#### 2.3.10. Estimation of Oxidative Stress Biomarkers

Hepatic tissue homogenates were used to spectrophotometrically determine lipid peroxidation in liver tissues with thiobarbituric acid-reactive substance (TBARS), and results were expressed as malondialdehyde (MDA) equivalents using 1,1,3,3 tetra methoxy propane as the standard [36]. Besides, superoxide dismutase (SOD) activity in liver tissues was estimated using the xanthine oxidase technique [37]. Reduced glutathione (GSH) was spectrophotometrically measured in liver tissues using Elman’s method [38]. Nitric oxide (NO) was spectrophotometrically assayed in liver tissues by measuring its stable metabolites, in particular, nitrite and nitrate [39].

#### 2.3.11. Estimation of Liver Biomarkers

Serum alpha fetoprotein-L3 (AFP-L3) levels were determined using an enzyme-linked immunosorbent assay kit from Glory Science (Hangzhou, China). Serum aspartate transaminase (AST) and alanine transaminase (ALT) levels were determined according to the method by Reitman and Frankel [40]. The activity of alkaline phosphatase (ALP) was estimated by the Belfield and Goldberg method [32]. Serum levels of total bilirubin were determined according to the technique by Walters and Gerarde [41].

#### 2.3.12. Histopathological Study

Tissues from rats’ livers were stored two days in buffered saline of formalin (10%), dried out by serial concentration of ethyl alcohol, cleared in xylene, inserted at 56 °C in paraffin for one day using a hot air oven. Staining the sections (4–5 μm) with the dye (Eosin and Hematoxylin (E&H)) and inspected beneath Olympus light microscope (Hicksville, NY, USA) [42].

#### 2.3.13. Statistical Analysis and Full Factorial Design Optimization

Statistical analysis, response surface methodology, and factorial optimization of the data were attained using MiniTab 17 software (San Diego, CA, USA). Data evaluations were completed using analysis of variance (ANOVA) followed by Tokay’s *t*-test [12,43]. Statistical significance was acceptable to a level of *p* < 0.05 and all relevant results were graphically displayed as mean ± SD.

## 3. Results and Discussion

### 3.1. Analytical Method Validation of LUT

The quantification of extracted active constituents is the most complex and demanding component of any plant extract. Different HPLC methods had previously been used to evaluate LUT as a flavonoid. For the determination of LUT concentration in our sample, we used a new simple spectrophotometric tool. Our method needs validation as a new technique of analysis. The maximum absorption peak (λ_max_) of LUT in EtOH: PBS, pH 7.4(1:3) was at 356 nm. The linearity of the LUT standard calibration curve ranged between 2–18 µg/mL (Appendix A). The statistical treatment of the data was carried out by using Linear regression analysis was conducted for statistical analysis of the data. The LOD and LOQ were 0.508 and 1.54 µg/mL, separately. Thus, our method has higher sensitivity than other previous methods. The results showed a close similarity between the measured and true results, thus exhibit excellent accuracy of the used methods (Appendix A). The average % of LUT recovery was close to 100% with low SD values, this indicates a higher accuracy of our method. These data demonstrated that our method is sensitive to a small change in drug concentration. The smallest value of relative SDs (<2%), demonstrating a high level of precision of our method in terms of inter-day and intra-day precision (Appendix A).

LUT solution (1:3 EtOH: PBS pH 7.4) was found to be stable for 3 days at 25 °C when kept away from light, and up to 10 days at 5 °C as it demonstrated no absorbance variations.

### 3.2. FTIR Spectroscopy

FTIR spectra of pure LUT, SL, EtOH, PG, cholesterol, physical mixture, and LUT loaded ethosomal formulation (F8) were analyzed to detect any interaction between LUT and any used excipients during the preparation of the ethosomal vesicles (Figure 2). The FTIR spectrum of Pure LUT showed characteristic peaks at 3405 and 1655 cm^−1^. The FTIR spectra of SL showed broadband 2920 and 1730 cm^−1^. The FTIR spectra of EtOH showed characteristic peaks at 33,910, 2952, and 1036 cm^−1^. The FTIR spectra of PG showed characteristic peaks at 3491, 1456, and 1109 cm^−1^. The FTIR spectra of cholesterol showed broadband at 3410.05 cm^−1^ and sharp broadband at 2934.66 and 1466.91 cm^−1^. The FTIR spectra of the physical mixture and LUT ethosomes showed that the characteristic peaks of LUT at 3405 cm^−1^ and 1655 cm^−1^ were not affected significantly. From the previous results, it was found that no chemical interaction has occurred either in the physical mixture or LUT-loaded ethosomes.

### 3.3. Ethosomes Morphology

The LUT-ENPs appeared under scanning electron microscopy in a nanosize scale with a round shape. The image demonstrates the efficiency of the cold method to develop the ethosomal vesicle (Figure 3).

### 3.4. Response Surface Methodology and Formulation Factors Optimization

#### 3.4.1. Entrapment Efficiency Percentage (EE%)

One of the essential dependent variables in the design of LUT-ENPs formulations is EE%. The independent variables were investigated and optimized by factorial design to achieve the highest EE%. Formula F8 gave the best EE% of the prepared LUT-ENPs, while formula F3 showed the worst value, as shown in Table 2.

Response surface plot (Figure 4a) showed that, as the concentration of SL increased, the EE% increased. EE% also increased with increasing the EtOH concentration from 15 to 30% then decreased as the EtOH concentration increased to 45% *v/v* as shown in Figure 4a. This may be due to the greater fluidity effect of EtOH which leads to LUT leakage from the prepared ethosomes. The relatively high entrapment of LUT within the vesicles (F8) is explained by the formation of multi-lamellar vesicles and the effect of both EtOH and SL content. These results agreed with previous data that showed that the EE% values for many hydrophobic drugs were improved due to the presence of EtOH in ethosomes [44].

#### 3.4.2. LUT-ENPs Size

Table 2 shows that the LUT-ENPs size significantly decreased by increasing ethyl alcohol concentration from 15% *v/v* to 30% *v/v*, due to the higher concentration of ethyl alcohol confers a surface negative net charge to the vesicular ethosomes by manipulating some surface characteristics, which causes the size of vesicles to decrease [45]. Response surface plots (Figure 4b) also showed that the further increase in ethyl alcohol concentration to 45% *v/v* increased the vesicle size which may be due to the provided negative charge by ethyl alcohol which led to a decrease in the net surface charge (zeta potential) that led to the agglomeration of the vesicles due to decreased electrostatic repulsion or rupture of the vesicles [46]. Moreover, the data showed an increase in ethosomal vesicle size by increasing SL concentration from 2% to 6% *w/v*. When the amount of SL was kept at 2% *w/v* and the concentration of ethyl alcohol was increased from 15% to 30% *v/v*, the size of the vesicles decreased from 355 ± 16.67 (F1) to 312 ± 6.98 nm (F2) Figure 4b. The same results were obtained when the amount of SL was kept at 4% w/v and 6% *w/v* and the ethyl alcohol concentration was increased in the same manner from 15% to 30% *v/v*, the vesicle size was decreased from 352 ± 9.56 nm (F4) to 292 ± 7.2 nm (F5) and 305 ± 10.3 nm (F7) to 267 ± 8.6 nm (F8) (Appendix A), respectively. This may be due to the reduction in the surface tension between the aqueous phase and the organic phase which resulted in a reduction in the ethosomal vesicle size [47]. The values of PDI showed that LUT-ENPs possess adequate homogeneity in particle size distribution (0.172 ± 0.05–0.463 ± 0.06 as shown in Table 2).

#### 3.4.3. Zeta Potential

The zeta potential represents the degree of vesicular system stability. Because of the repulsive force between the charged vesicles, higher zeta potential values mean that the particle has little capacity to aggregate. Formulations with a zeta potential equal to or greater than 30 mV are classified as stable preparations [48]. The zeta potential of the prepared LUT-ENPs formulations was ranged between −30.1 ± 1.2 and −42.6 ± 3.01 mV as shown in Table 2, Figure 4d, and Appendix A, thus the prepared systems could possess adequate stability.

#### 3.4.4. In Vitro Release of LUT from ENPs Formulations

Table 2, Figure 4c and Figure 5 show the LUT in vitro release profile from LUT-ENPs. In contrast to the aqueous LUT suspension, both ethosomal preparations showed a substantial increase in dissolution rate after 12 h. The results showed that increasing the EtOH concentration increased the in vitro release of LUT from ethosomes, which may be attributed to increased fluidity of the ethosomal vesicles bilayer membrane with increasing the EtOH concentration [49]. Response surface plot (Figure 4c) shows a significant decrease in the release profile of LUT when the SL concentration was increased from 2% to 6% *w/v*, with constant EtOH concentration. This result could be attributed to the fusion of the vesicles together resulting in larger vesicles [50].

### 3.5. Kinetic Study

The kinetic results obtained from the in vitro release of LUT from LUT-ENPs preparations (F1–F9) were best described by the Higuchi-diffusion model based on the highest r^2^ values (Appendix A).

### 3.6. Selection of the Best LUT-ENPs Formula

The best LUT loaded ethosomal formulation was selected depending on the highest EE%, lowest vesicle size, highest zeta potential value, and appropriate % of LUT released after 12 h. depending on this parameter, F8 (containing 30% *v/v* EtOH, 5% *w/v* SL, 1% cholesterol, and 10% *v/v* PG) were selected as the best formula to be further investigated.

### 3.7. LUT-ENPs Stability Study

The stability studies of LUT-ENPs (F8) are shown in Figure 6. The amount of LUT leached from ethosomes was increased as the storage time increased. This phenomenon correlated with the inability of the small vesicles to encapsulate LUT for a long time [51]. Significantly higher drug leaching occurs at elevated temperatures due to the higher fluidization of the vesicular bilayer [50]. The size of the stored ethosomes also increased as the storage time increased, this was due to aggregation of the vesicles and loss of the spherical shape of vesicles at high temperature [52]. Moreover, the % of LUT released was increased in accordance with the increase of the storage time. The vesicle fluidity at room temperature was increased which provided higher leaching (decreased EE%) and rapid release of LUT from the vesicles [53].

### 3.8. In Vivo Study

#### 3.8.1. Gene Expression Levels of GPC3 by RT-PCR

The results proved that the hepatic GPC3 gene expression was significantly increased in carcinogenic groups (*p* < 0.05) with respect to control groups. These results were matched with the study which reported that GPC-3 overexpressed in HCC, and its expression level serves as a promising prognostic biomarker. Additionally, GPC3 may also be a hopeful molecular target for the improvement of innovative therapies to enhance the prognosis of HCC patients [54]. Treatment with LUT suspension does not affect the hepatic GPC3 gene expression. After treatment with LUT-ENPs, the hepatic GPC3 gene expression was significantly decreased in comparison to carcinogenic groups (*p* < 0.05) [55,56] (Figure 7).

#### 3.8.2. Serum Levels of, ALT, AST, ALP, Total Bilirubin, and AFP-L3

By comparing the data obtained from carcinogenic groups and healthy groups we found that liver biomarkers, total bilirubin, and AFP-L3 were significantly elevated in the carcinogenic groups (*p* < 0.05) with respect to healthy groups. Liver enzymes, (ALT, AST, and ALP) are reliable markers to determine liver activity [57]. When the liver is damaged or injured, a breakdown in the cell membrane architecture is likely to happen which leads to the release of these enzymes into the serum. Furthermore, AFP-L3 is a serological marker for the diagnosis of HCC with significant-high sensitivity [58]. After treatment with LUT suspension, there is no significant effect on the liver biomarker, ALP, total bilirubin, and AFP-L3. On the other hand, after treatment with LUT-ENPs, there was a significant decrease in liver biomarkers, total bilirubin, and AFP-L3 in comparison to carcinogenic groups (*p* < 0.05) as shown in Figure 8. These results agreed with other previous findings [59,60].

#### 3.8.3. Hepatic GSH, SOD, NO, and MDA Content

The results showed that the hepatic tissue levels of GSH and SOD were significantly increased in the group treated with LUT-ENPs with respect to diseased groups (*p* < 0.05). These results were in coordination with findings of Liu et al. [59] who reported that there was a significant decrease in hepatic tissue content of both GSH and SOD in carcinogenic groups in comparison to the control group while treatment with natural extract as rhizoma paridis showed a significant increase in hepatic tissue content of GSH and SOD compared to DENA group. Furthermore, these results were in the same line with the study which reported that both oxidative stress and free radicals are one of the main factors in the progression of cancer [61]

Moreover, the result showed a significant decrease in the hepatic tissue NO and MDA content in the group treated with LUT-ENPs in comparison to carcinogenic groups. These results were in coordination with findings of Liu et al. [62] who reported that there was a significant increase in hepatic tissue content of both MDA and NO in the carcinogenic group in comparison to the control group while treatment with natural extract as rhizoma paridis showed a significant decrease in hepatic tissue content of both MDA and NO compared to DENA group

Besides, hepatic tissue NO and MDA content significantly decreased in the group treated with LUT-ENPs in comparison to the group treated with LUT suspension [61,62] (*p* < 0.01). (Figure 9).

#### 3.8.4. Histopathological Examination

The collected animal livers’ histopathological characteristics revealed that the control groups had normal hepatocytes arranged in ropes around the central vein, while the diseased groups had HCC knobs. These HCC knobs remained visible after treatment with LUT suspension. On the other hand, the treatment with LUT-ENPs (F8) appeared hepatic foci with a significant diminish of neoplastic hepatic injuries with a little number of hepatic adenomas as appeared in Figure 10.

## 4. Conclusions

Several batches of LUT-ENPs were successfully prepared. The concentration of EtOH and SL used had a major impact on EE percent, vesicle size, and in vitro drug release. LUT-ENPs offers a long-term LUT release profile. LUT-ENPs in the nano-size range were spherical and had a high EE percent. The zeta potential of the prepared LUT-ENPs indicated that the system is stable. LUT-ENPs (F8) was selected as the best formula and exposed to several examinations as FTIR that was shown no interaction between the drug and excipients. The results of stability studies showed that the amount of LUT leached from LUT-ENPs, the vesicle size, and the percent of LUT released were increased as the storage time increased. Significant improvement of liver biomarkers, decreased expression of GPC3 gene and the hepatic tissue NO and MDA content, and increased hepatic tissue levels of GSH and SOD were observed with LUT-ENPs. Histopathological examination showed a small number of hepatic adenomas and a significant decrease of neoplastic hepatic lesions for the tissues treated with LUT-ENPs. It was concluded that LUT-ENPs is a promising oral drug delivery system for enhancing the anti-tumor activity of LUT against HCC.

## Figures and Tables

**Figure 1 pharmaceutics-13-00648-f001:**
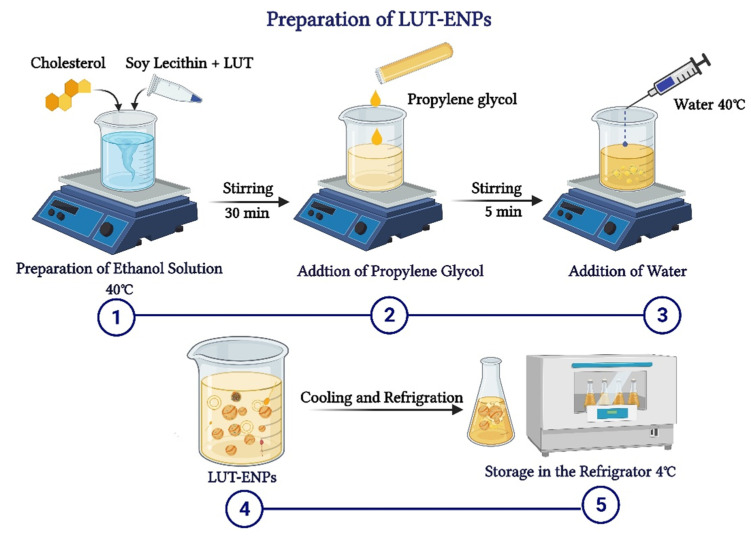
Schematic diagram showing the detailed cold method steps for preparation of luteolin-loaded ethosomal nanoparticles (LUT-ENPs).

**Figure 2 pharmaceutics-13-00648-f002:**
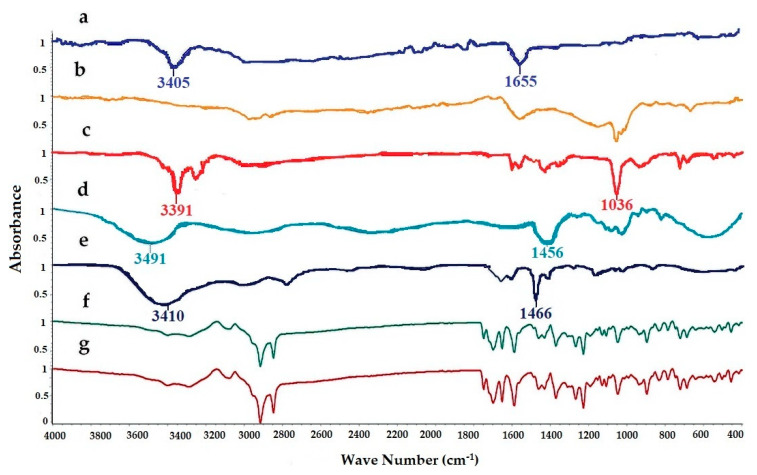
FTIR spectra of: (**a**) luteolin (LUT), (**b**) SL, (**c**) ethanol (EtOH), (**d**) propylene glycol (PG), (**e**) cholesterol, (**f**) physical mixture, and (**g**) LUT-ENPs (F8).

**Figure 3 pharmaceutics-13-00648-f003:**
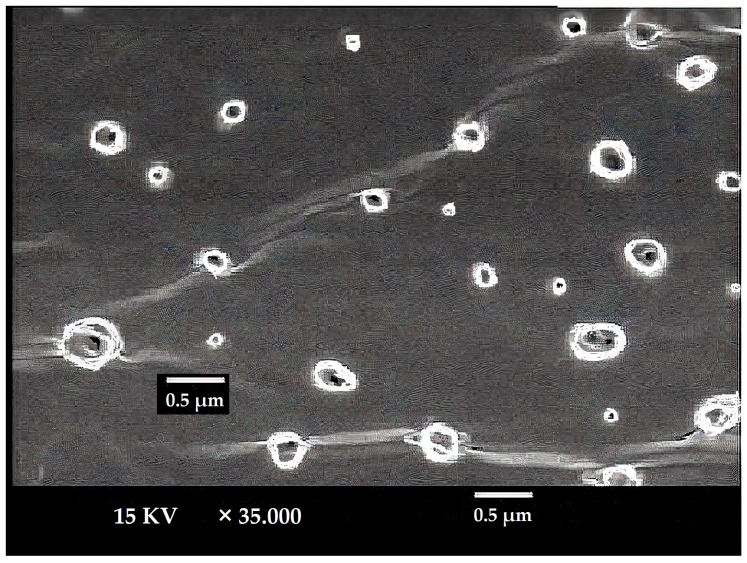
SEM micrograph of selected LUT ethosomes (F8).

**Figure 4 pharmaceutics-13-00648-f004:**
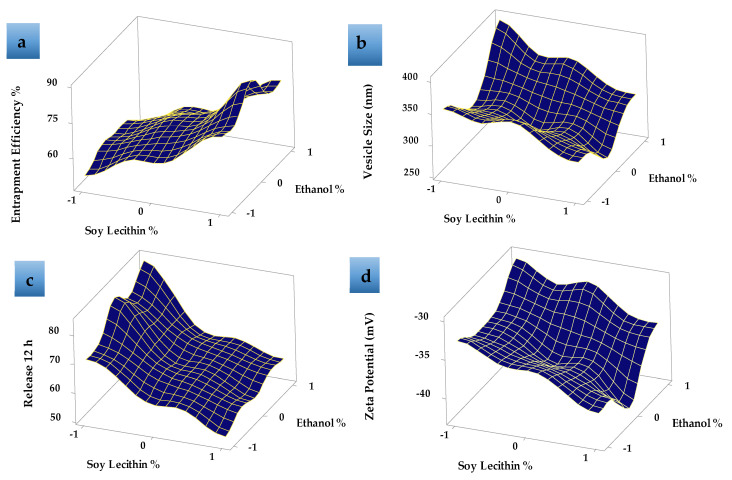
Response surface plots effects of different formulation factors on (**a**) entrapment efficiency %, (**b**) vesicle size (nm), (**c**) release 12 h, and (**d**) zeta potential (mV).

**Figure 5 pharmaceutics-13-00648-f005:**
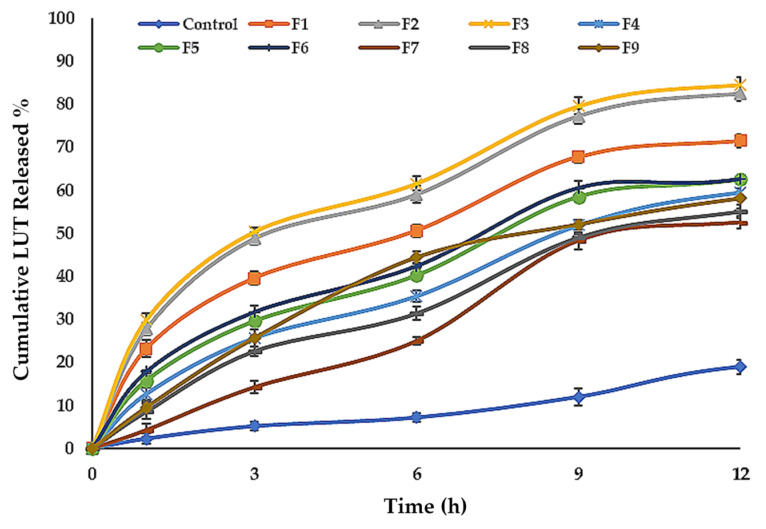
Cumulative percentage release of LUT from ethosomal formulations (mean ± SD, *n* = 3). Control was an aqueous suspension of the drug.

**Figure 6 pharmaceutics-13-00648-f006:**
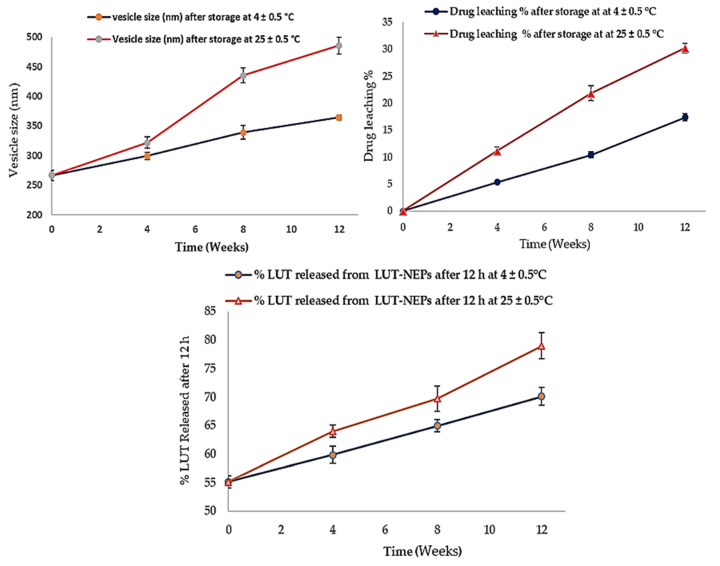
Stability profiles of the selected LUT-ENPs (F8) under storage at 4 °C and 25 °C for 12 weeks (mean ± S.D, *n* = 3).

**Figure 7 pharmaceutics-13-00648-f007:**
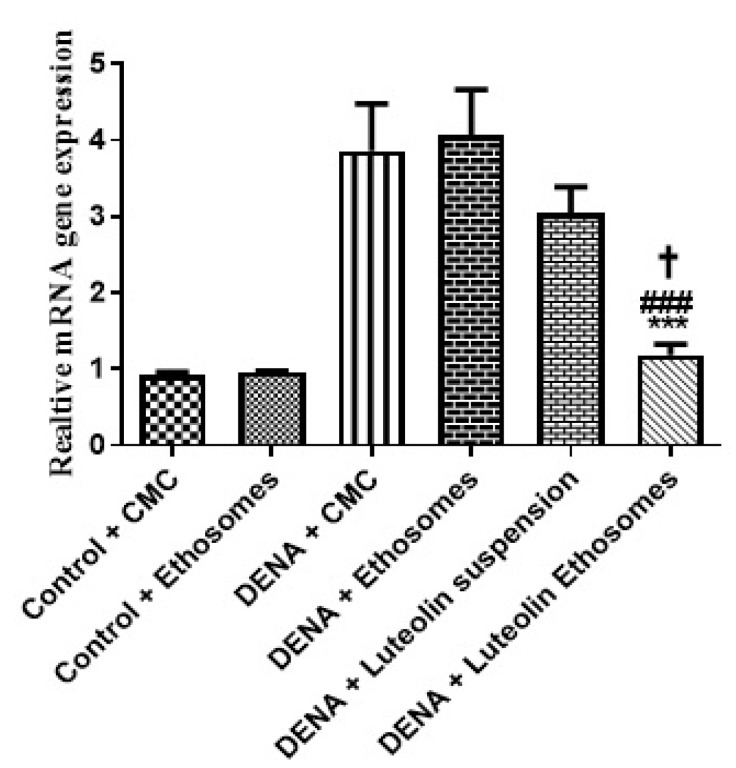
Effects of LUT and/or LUT-ENPs on gene expression of GPC3. Data are presented as mean ± SEM (*n* = 12). *, #, and † indicate significant changes from control, DENA, and DENA+ luteolin suspension groups, respectively. † indicates significant change at *p* < 0.05; *** and ### indicate significant change at *p* < 0.001.

**Figure 8 pharmaceutics-13-00648-f008:**
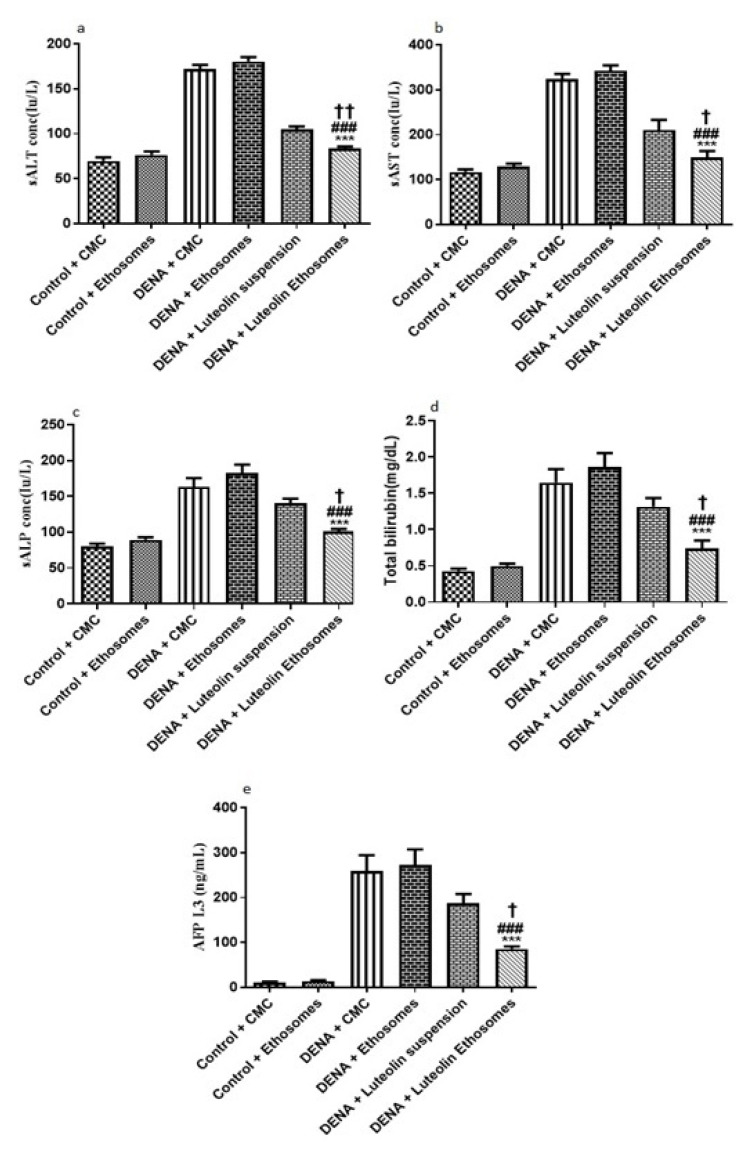
Effects of LUT and/or LUT-ENPs on (**a**) alanine transaminase (ALT), (**b**) aspartate transaminase (AST), (**c**) alkaline phosphatase (ALP), (**d**) total bilirubin, and (**e**) alpha fetoprotein-L3 (AFP-L3) Data are presented as mean ± SEM (*n* = 12). *, #, and † indicate significant changes from control, DENA, and DENA+ Luteolin suspension groups respectively. † indicates significant change at *p* < 0.05; †† indicates significant change at *p* < 0.01; *** and ### indicate significant change at *p* < 0.001.

**Figure 9 pharmaceutics-13-00648-f009:**
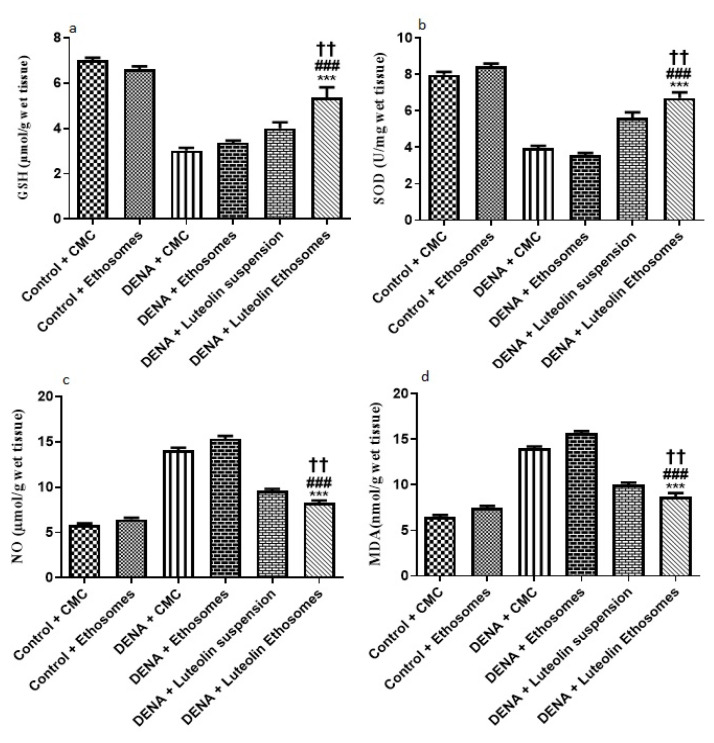
Effects of LUT and/or LUT-ENPs on, (**a**) glutathione (GSH), (**b**) superoxide dismutase (SOD), (**c**) nitric oxide (NO), and (**d**) malondialdehyde (MDA). Data are presented as mean ± SEM (*n* = 12). *, # and † indicate significant changes from control, DENA, and DENA+ Luteolin suspension groups respectively. †† indicates significant change at *p* < 0.01; *** and ### indicate significant change at *p* < 0.001.

**Figure 10 pharmaceutics-13-00648-f010:**
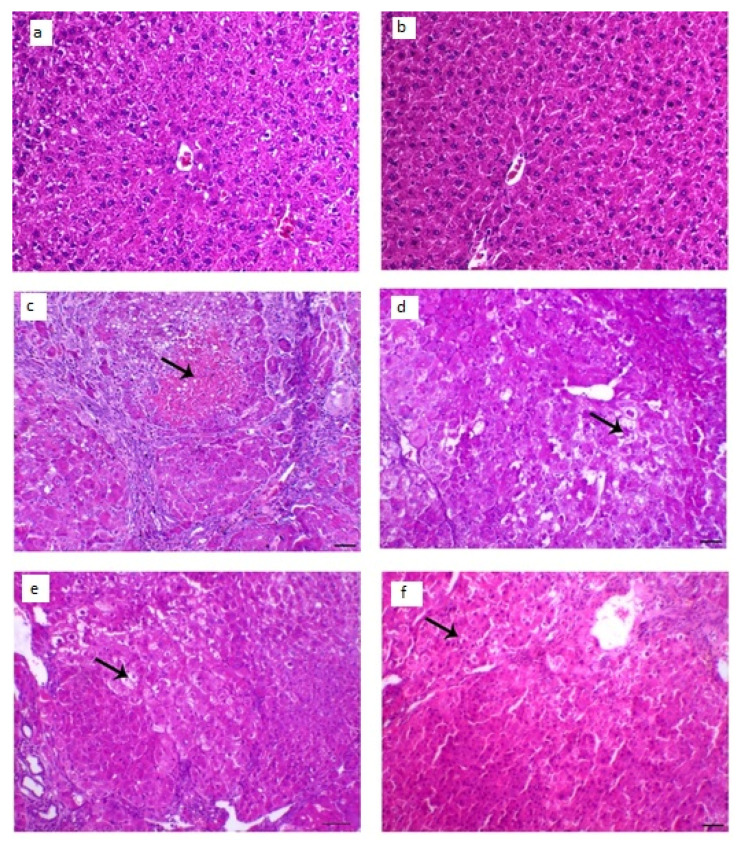
Histopathological changes in liver tissues; (**a**,**b**) liver of control animal appearing typical hepatocytes organized in lines around the central vein (arrow; H&E staining; scale bar, 100 µm, (**c**,**d**) liver section of the diseased animal showing HCC nodule revealing marked hepatic necrosis (arrow), H&E, bar = 100 µm., (**e**) liver segment of infected animal treated with LUT-suspension showing hepatic adenoma with hepatic vacuolar degenerative changes (arrow), H&E, bar = 100 µm, (**f**) liver segment of infected animal treated with LUT-ENPs appearing hepatic foci with marked diminish the hepatic neoplastic injuries with a little number of hepatic adenomas (arrow shows central degenerative zone), H&E, bar = 100 µm.

**Table 1 pharmaceutics-13-00648-t001:** Composition of different LUT-ENPs formulations.

Composition	F1	F2	F3	F4	F5	F6	F7	F8	F9
Luteolin (mg)	10	10	10	10	10	10	10	10	10
Ethanol (% *v/v*) (X_1_)	15	30	45	15	30	45	15	30	45
Soy Lethicin (% *w/v*) (X_2_)	2	2	2	4	4	4	6	6	6
Cholesterol (% *w/v*)	1	1	1	1	1	1	1	1	1
PG (% *v/v*)	10	10	10	10	10	10	10	10	10
Water	Q.S.	Q.S.	Q.S.	Q.S.	Q.S.	Q.S.	Q.S.	Q.S.	Q.S.

Ethanol % level code: 15% (−1), 30% (0), and 45% (+1); Soy Lethicin % level code: 2% (−1), 4% (0), and 6% (+1).

**Table 2 pharmaceutics-13-00648-t002:** Different dependent formulation parameters of LUT-ENPs.

F. Code	EE%	Vesicle Size (nm)	PDI	Zeta Potential (mV)	Cumulative % Released at 12 h
F1	52.12 ± 0.08	355 ± 16.67	0.458 ± 0.04	−32.8 ± 1.3	71.48± 1.34
F2	57.72 ± 0.35	312 ± 6.98	0.463 ± 0.06	−34.4 ± 1.43	82.47 ± 2.03
F3	49.2 ± 2.1	401 ± 4.76	0. 399 ± 0.03	−30.1 ± 1.2	84.47 ± 1.37
F4	62.35 ± 0.31	352 ± 9.56	0.218 ± 0.02	−35.2 ± 2.01	59.53 ± 1.09
F5	67.9 ± 0.45	292 ±7.8	0.344 ± 0.05	−38.3 ± 1.05	62.50 ± 2.34
F6	58.6 ± 1.23	359 ± 5.43	0.321 ± 0.01	−31.7 ± 2.32	62.55 ± 1.78
F7	78.66 ± 0.66	305 ± 10.3	0.317 ± 0.03	−39.2 ± 2.23	52.52 ± 1.93
F8	89.77 ± 0.86	267 ± 8.6	0.172 ± 0.05	−42.6 ± 3.01	55.11 ± 1.07
F9	76.66 ± 1.22	319 ± 7.6	0.23 ± 0.09	−35.5 ± 2.9	58.12 ± 2.32

## Data Availability

Not applicable.

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
