# Peer review of "Design and Optimization of Orally Administered Luteolin Nanoethosomes to Enhance Its Anti-Tumor Activity against Hepatocellular Carcinoma"

_pharmaceutics, 2021, doi:10.3390/pharmaceutics13050648_

Round 1
Reviewer 1 Report
Elsayed et al. demonstrate the use of ethosomes to enhance the oral bioavailability of a natural flavanoid, Luteolin, for the treatment of hepatocellular carcinoma. Overall, the authors employed a response surface methodology to optimize the formulation and use the optimized formulation for treatment of chemical induced hepatocellular carcinoma. While, the results, especially the in vivo efficacy results are interesting, a few comments must be addressed before the manuscript is considered for publication.
- Authors should explicitly describe their design of experiment strategy in the methods section. Specifically a table describing all the factors, their levels and the measured outcomes would greatly benefit the reader's understanding.
- While the analytical method development and validation is critical for natural compounds, authors should consider moving Fig 2 to supplemental information as it diverts from the main aspect of this manuscript which is physicochemical and functional characterization of LUT-ENPs.
- In FTIR spectra of all the excipients (Fig 3), the spectra of LUT seems to be unclear. Based on the absorbance values, the characteristic peaks mentioned by authors(line 291) are not explicitly evident. Authors should provide a more clear FTIR spectra or carefully draw inferences form this particular one. Based on the current spectra, it would be incorrect to draw conclusions about the characteristic peaks.
- Authors state that by increasing the EtOH concentration from 15 to 30%, they observed an increase in %EE, but at the same shift in EtOH concentration, the observed a decrease in particle size. While the authors attribute this to the change in surface properties, it is still counter intuitive to see the decrease after an increase in the drug loading. Authors should clarify this further. The unit for EtOH concentration in Table 1 is mentioned as % w/v while % v/v is used in the intermitently in the manuscript. Authors should use a consistent unit throughout.
- Authors performed in vitro release study under neutral buffer conditions. Authors must validate the effect of gastric pH on release and stability of LUT-ENPs , considering oral bioavailability to be the key aspect of this study.
- How did the PDI of F8 change on storage?
- Lastly, authors should demonstrate why the LUT-ENPs are outperforming the LUT suspension. Is this an effect of improved tissue accumulation or an increased half-life of LUT in the blood? Blood/ tissue concentration/ bioavailability should be reported to explain the downstream functional effect of LUT-ENPs.
Author Response
We appreciate the reviewer for your valuable and constructive comments and suggestions, which greatly helped us to improve the manuscript. In the light of your comments, we revised our manuscript and respond to these valuable comments as follows:
- Authors should explicitly describe their design of experiment strategy in the methods section. Specifically, a table describing all the factors, their levels, and the measured outcomes would greatly benefit the reader's understanding.
Response:
Thanks for the valuable suggestion. We added the required data to the manuscript page 3 line 132-135 and line 141 under Table 1 describes all factors levels, whereas measured outcomes were already illustrated in Table 2
- While the analytical method development and validation is critical for natural compounds, authors should consider moving Fig 2 to supplemental information as it diverts from the main aspect of this manuscript which is the physicochemical and functional characterization of LUT-ENPs.
Response:
We are also with this suggestion, and the figure was removed from the manuscript and added to the supplementary file (figure 1)
- In FTIR spectra of all the excipients (Fig 3), the spectra of LUT seem to be unclear. Based on the absorbance values, the characteristic peaks mentioned by the authors (line 291) are not explicitly evident. Authors should provide clearer FTIR spectra or carefully draw inferences form this particular one. Based on the current spectra, it would be incorrect to draw conclusions about the characteristic peaks.
Response:
We edit FTIR spectra in clear form and all peaks are now explicitly evident and all significant peaks were labeled as shown on page 8 line 305
- The authors state that by increasing the EtOH concentration from 15 to 30%, they observed an increase in EE%, but at the same shift in EtOH concentration, the observed a decrease in particle size. While the authors attribute this to the change in surface properties, it is still counter intuitive to see the decrease after an increase in the drug loading. Authors should clarify this further. The unit for EtOH concentration in Table 1 is mentioned as % w/v while % v/v is used in the intermitently in the manuscript. Authors should use a consistent unit throughout.
Response:
- BY increasing the EtOH concentration from 15 to 30%, %EE was increased since at this level ethanol increases the solubility of luteolin and increases drug loading. (1). the LUT-ENPs size significantly decreased by increasing ethyl alcohol concentration from 15% v/v to 30 % v/v, due to the higher concentration of ethyl alcohol confers a surface negative net charge to the vesicular ethosomes by manipulating some surface characteristics, which causes the size of vesicles to decrease (2). The same result obtained with El –Shenawy et al(3)
- We corrected the unit for EtOH concentration to %v/v.
5- Authors performed in vitro release study under neutral buffer conditions. Authors must validate the effect of gastric pH on release and stability of LUT-ENPs, considering oral bioavailability to be the key aspect of this study.
Response:
We appreciate the reviewer for this valuable comment. In fact, in our preliminary trials we used the gastric pH (1.2) in both in vitro release and stability, we founded that there is a non-significant difference between it and neutral buffer. So, we used buffer throughout our study.
- How did the PDI of F8 change on storage?
Response:
The size of the stored ethosomes also increased as the storage time increase, due to aggregation of the vesicles and loss of the spherical shape of vesicles at high temperature (4)
- Lastly, the authors should demonstrate why the LUT-ENPs are outperforming the LUT suspension. Is this an effect of improved tissue accumulation or an increased half-life of LUT in the blood? Blood/ tissue concentration/ bioavailability should be reported to explain the downstream functional effect of LUT-ENPs.
Response:
The LUT-ENPs are outperforming the LUT suspension due to the ability of LUT-ENPs to provide controlled release of the drug so it increases the half-life of LUT in blood and also reduces the rate of clearance. In addition, the encapsulation of LUT within the ethosomal vesicles protects it from degradation thus improving the efficiency of the drugs (5).
1 Abdulbaqi IM, Darwis Y, Khan NAK, Abou Assi R , Khan AAJIjon. Ethosomal nanocarriers: the impact of constituents and formulation techniques on ethosomal properties, in vivo studies, and clinical trials. (2016). 11, 2279
2 Dubey V, Mishra D, Dutta T, Nahar M, Saraf D , Jain N. Dermal and transdermal delivery of an anti-psoriatic agent via ethanolic liposomes. J Control Release (2007). 123, 148-154
3 El-Shenawy AA, Abdelhafez WA, Ismail A , Kassem AA. Formulation and Characterization of Nanosized Ethosomal Formulations of Antigout Model Drug (Febuxostat) Prepared by Cold Method: In Vitro/Ex Vivo and In Vivo Assessment. AAPS PharmSciTech (2019). 21, 31-31, doi:10.1208/s12249-019-1556-z
4 Haruyama Y ,Kataoka H. Glypican-3 is a prognostic factor and an immunotherapeutic target in hepatocellular carcinoma. World journal of gastroenterology (2016). 22, 275-283, doi:10.3748/wjg.v22.i1.275
5 Sankhyan A ,Pawar P. Recent trends in niosome as vesicular drug delivery system. J Appl Pharm Sci (2012). 2, 20-32
Reviewer 2 Report
In this research, authors developed ethosomal nanoparticles (ENPs) as a drug procurement route for LUT to increase its bioavailability and anti-cancer activity. The ENPs were characterized by abundant methodology including in vitro and in vivo models.I suggest to accept the manuscript for publication, after authors address the following the questions.
1) It is better to label the characteristic peaks in the FTIR spectra.
2) In figure 4, why F4 has distictive release profile?
3) For figure 7b, it is better to change curve color at 25 degree c to red.
Author Response
We appreciate the reviewer for your valuable and constructive comments and suggestions, which greatly helped us to improve the manuscript. In the light of your comments, we revised our manuscript as follows:
1) It is better to label the characteristic peaks in the FTIR spectra.
Response:
We edit FTIR spectra in clear form and all characteristic peaks were labeled as shown in page 8 line 305
2) In figure 4, why F4 has distictive release profile?
Response:
Since it composed of a low level of ethanol (15%v/v) and a medium level of soy lecithin (4% w/v) which provide a distinctive release profile
3) For figure 7b, it is better to change the curve color at 25 °C to red.
Response:
The figure 7b, the curve color at 25 °C was changed to red.
(Note: All figures numbers were changed after removal of figure 2 and moving it to supplementary materials)
Round 2
Reviewer 1 Report
Elsayed et al. have addressed each of the comments and have submitted an improved manuscript. I am satisfied with the revised version.